# Moderating Roles of Social Support in the Association between Hope and Life Satisfaction among Ethnic Minority College Students in China

**DOI:** 10.3390/ijerph18052298

**Published:** 2021-02-26

**Authors:** Xin Chen, Yonghong Ma, Ruilin Wu, Xia Liu

**Affiliations:** 1School of Humanities and Social Science (School of Public Administration), Beihang University, No. 37 Xueyuan Road, Haidian District, Beijing 100191, China; cx@buaa.edu.cn (X.C.); myhong@buaa.edu.cn (Y.M.); wuruilin@buaa.edu.cn (R.W.); 2Institute of Developmental Psychology, Beijing Normal University, No. 19 Xinjiekouwai Street, Beijing 100875, China

**Keywords:** hope, Han social support, minority social support, life satisfaction, ethnic minority college students

## Abstract

Ethnic minority college students in China are Chinese students who migrate from ethnic minority-dominant areas to Han-dominant areas to attend college. Acculturative stress would lead to maladjustments for ethnic minority college students in China, such as low levels of life satisfaction. To help improve the life satisfaction of these students, this study adopted resilience theory to explore the beneficial effects of hope and social support and their influencing mechanisms. The participants included 362 ethnic minority college students in Beijing, China, and a questionnaire method was used. The results indicated that Han social support had a promotive effect on life satisfaction, while hope did not have a promotive effect on life satisfaction. Moreover, Han social support moderated the relationship between hope and life satisfaction. Specifically, the effect of hope on life satisfaction was stronger under the condition of a higher level of Han social support compared with those under a lower level of Han social support. In conclusion, Han social support can play a promotive effect individually, while hope only affected life satisfaction when a high level of Han social support existed. Han social support should be strengthened to improve the life satisfaction among ethnic minority college students in China.

## 1. Introduction

Ethnic minority college students in China are sojourning students who migrate between different subcultures within China [1]. China has been inhabited by people from diverse ethnic backgrounds for many centuries. The central government currently recognizes 56 different nationalities or ethnic groups within China. These ethnic minority groups have their own ethnic cultures that are distinct from the dominant Han culture. All but two (Hui and Manchu), for example, have their own languages, and most have their own religious beliefs (e.g., Tibetans are followers of Buddhism). Ethnic minority groups also typically have their own dietary habits and modes of dress that differ from Han culture. The Hui, for example, follow Islamic principles regarding food preparation. Such cultural practices can give rise to significant dietary differences between Han and other ethnic minority groups [1].

Economic development in ethnic-minority areas generally lags that in predominantly Han regions. The government has implemented various policies in recent years to accelerate development in such areas. One important aim of these policies, according to the National Long-Term Education Reform and Development Plan (2010–2020), is to enroll more ethnic minority students in large Chinese universities [2]. As a result, large numbers of ethnic minority students now attend college in Han-dominant regions. According to the Ministry of Education of China, 1,263,936 ethnic minority students studied in Han-dominant Chinese universities in 2014, accounting for 7.92% of all Chinese college students. Even though such students migrate within their own country, they still confront cultural shock when they migrate to Han regions [3,4,5]. For example, dietary habits are one of the challenges confronting ethnic minority college students [3]. In addition, language barriers also exist for ethnic minority college students. Although most of them are fluent in Mandarin in daily life, specialized words in majors are still difficult for them to understand [4], and some students feel that they are rejected by Han people for their strong accent in Mandarin [5]. These acculturative stresses lead ethnic minority college students to maladjustments, such as low level of life satisfaction [6,7]. Life satisfaction refers to, more generally, enduring background appraisals encompassing one’s overall life or major facets of one’s life [8]. Studies have found that ethnic minority college students in China have lower life satisfaction than Han college students [6,7].

Yet, little research has focused on exploring the protective factors and influencing mechanisms related to the life satisfaction of these students. This study aimed to fill this knowledge gap by investigating the effects of hope and social support, and the moderating role of social support, on the relationship between hope and life satisfaction among ethnic minority college students in China. Specifically, adopting the protective–protective model in resilience theory, we explored whether Han social support and minority social support amplified or attenuated the effect of hope on life satisfaction. This work can provide culturally specific evidence for broadening the foundation of resilience theory. Furthermore, this research can provide a basis for expanding related empirical research while also developing culturally competent interventions to improve life satisfaction among ethnic minority college students in China.

### 1.1. Hope, Social Support, and Life Satisfaction

Hope is a trait-like disposition comprising two characteristic cognitions regarding agency and pathways [9]. Many studies have found that hope is positively associated with life satisfaction among different cultures and groups. For instance, O’Sullivan found that, compared to eustress and self-efficacy, hope was a better predictor of life satisfaction among students at Pitzer College, such that an increase in hope could enhance their life satisfaction [10]. Likewise, Hoy found that hope was positively associated with life satisfaction among elementary school students in the US, where an increase in hope would lead to higher life satisfaction [11]. Given the lack of related research in the Chinese context, the first aim of the present study was to explore the relation between hope and life satisfaction for ethnic minority college students in China.

Social support refers to a person’s sense that he or she is cared for, loved, and respected and that he or she belongs to a network characterized by mutual obligation [12]. Social support has been found to provide valuable resources for coping with stress and maintaining good health and well-being (including life satisfaction) in different cultures and groups [13,14]. Acculturation research tends to divide migrants’ social support into two types: local and nonlocal. Local can include people from the host culture, while nonlocals can include people from the migrant’s own culture [15]. It has been suggested that local social support always has a promotive effect [16]. The effects of nonlocal support, however, have varied across different studies. Noh and Kaspar found that among refugees who had spent an average of 20 years in the host country, those who were not sufficiently supported by their own cultural communities experienced more negative psychological effects related to acculturation [17]. However, Jasinskaja-Lahti and Liebkind, studying immigrants from the former USSR residing in Finland, found that too much nonlocal social support could harm well-being, increase discrimination, and hinder adaptation [18].

Meanwhile, the effects of local and nonlocal social support on the life satisfaction of ethnic minority college students in China remain unknown. Shedding light on this issue can help provide new insights for carrying out targeted interventions to improve the life satisfaction of this disadvantaged group. Since Han social support is the kind of social support provided by Han people, it is equal to local support in Han regions, and minority support is equal to nonlocal social support in the same way. The second aim of this research was to clarify the relations among Han social support, minority social support, and life satisfaction for ethnic minority college students in China.

### 1.2. Moderating Role of Social Support in the Relation between Hope and Life Satisfaction

Resilience theory suggests that the effects of promotive factors are not universal—that is, they can vary regarding different groups, contexts, and outcomes [19]. Currently, there is little empirical research on the effects and influencing mechanisms of hope and social support on life satisfaction among ethnic minority college students in China. 

Fergus and Zimmerman proposed a protective–protective model in the context of resilience theory to describe the influencing mechanism of two protective factors [19]. They argued that the protective–protective model only applies to resilience when the two protective factors are studied for a disadvantaged group. They also explained that the two protective factors have an interaction effect in producing an outcome. Based on this theoretical framework, hope and social support might have an interaction effect regarding life satisfaction among ethnic minority college students in China, who have been proven as a disadvantaged group by many studies [3,4,5]. Which factor would be the moderator—hope or social support? In previous studies, social support, as an important resource factor, is usually found to play a moderating role in the relations between personal factors and developmental results. Noh and Kaspar, for example, found that the coping efficacy of Korean immigrants in Toronto was determined by social support [17]. Studying mainland Chinese university students who migrated to Hong Kong, Ng et al. found that social support moderated the effect of acculturation strategies on cultural adaptation [15]. Accordingly, the effect of hope—as a personal or asset factor—on life satisfaction may be moderated by social support.

Hope theory and prior studies provide further support for the above inference, suggesting that social support has an enhancing effect on hope. Snyder, for example, suggested that social support can improve the level of hope by building and broadening resources [9]. The extended hope theory proposed by Bernado further emphasizes the enhancing effect of social support on hope, especially in collectivistic cultures [20]. Prior empirical studies have also found that social support can positively affect hope [21,22]. Based on the above, we inferred that social support could have a moderating effect on the relation between hope and life satisfaction among ethnic minority college students in China.

Additional research using the protective–protective model has specified two types of moderating models [23,24]. One is the enhancing model whereby a protective factor can enhance the effect of another promotive factor in producing an outcome. The other is the antagonistic model whereby a protective factor can decrease the effect of another promotive factor in producing an outcome. These two moderating models serve to further clarify the moderating effect and the orientation of the intervention.

The third aim of this research, then, was to test the moderating role of Han and minority social support in the relationship between hope and life satisfaction. If the enhancing model is supported, related interventions should focus on increasing social support to improve the life satisfaction of ethnic minority college students in China. If the antagonistic model is supported, interventions should focus on increasing hope.

In addition, many studies about ethnic minority college students in China proved that gender differences existed regarding hope and social support [25,26]. For example, Li and He’s study showed male students had a higher level of hope than female students [25]. Another study about ethnic minority college students showed female students perceived more social support than male students [26]. Based on the gender differences on those variables, our fourth research aim was to examine whether the above interaction model would have gender differences.

### 1.3. The Current Study

This study had four aims. The first aim was to explore the relationship between hope and life satisfaction among minority college students in China. The second aim was to investigate the effect of Han social support and minority social support on the life satisfaction of that group. The third aim was to test the moderating role of Han social support and minority social support in the relation between hope and life satisfaction. The fourth aim was to examine whether the interaction model would show differences between the two groups of ethnic minority college students (i.e., female vs. male).

## 2. Materials and Methods

### 2.1. Participants

Using convenience sampling, we recruited 362 ethnic minority students from four universities in Beijing. Participants were all freshmen. During the process of recruiting, we contacted the students’ advisors in four universities to help us recruit participants. Those students who were interested in our study were enrolled. As shown in Table 1, the participants consisted of 156 males (42.8%) and 206 females (57.2%). A total of 227 attended comprehensive colleges (63.2%), and 135 attended ethnic minority colleges (36.8%). Thirty-eight participants came from large cities (10.6%), 53 from medium-sized cities (14.8%), 184 from small towns (50.8%), and 87 from villages (23%). Twenty-two ethnic minority groups were represented in the sample, including China’s 10 largest minority groups: Zhuang, Hui, Manchu, Uyghur, Miao, Yi, Tujia, Tibetan, Mongol, and Dong. All participants were fluent in Mandarin Chinese.

### 2.2. Procedure

The survey was conducted one month after enrollment. This decision was based on prior research indicating that the first month after arrival can be the most difficult time in the cultural adaptation process [27]. 

Before conducting the survey, we aimed to ensure the rights of ethnic minority college students and avoid any perceived threats arising from cultural differences. First, we let the students’ advisers, who are very familiar with the cultural customs of minorities, examine the questionnaire. Then, we chose a few ethnic minority students to conduct a pretest. On that basis, we removed some language perceived as sensitive and changed descriptions students had difficulty understanding. All study procedures were approved by the Institutional Review Board (IRB) of BeiHang University.

Surveys were conducted by contacting class advisers at the four universities. After obtaining permission from the adviser, we distributed questionnaires during class meeting times. The questionnaires took approximately 15 min to complete. During the survey process, we first obtained informed consent from each student and assured them they could withdraw from the study at any time. After the surveys, we sent a pen worth approximately 10 RMB to each participant.

### 2.3. Measurements

#### 2.3.1. Hope

Hope was measured using the Dispositional Hope Scale developed by Feldman and Snyder [28]. This scale contains 12 items: 4 items assess agency (e.g., I energetically pursue my goals), 4 assess pathways (e.g., I can think of many ways to get out of a jam), and 4 are distracters (e.g., I worry about my health). Each item was rated on a 4-point scale, ranging from 1 (definitely false) to 4 (definitely true). Agency and pathway subscale items were summed to yield a total hope score. The distracter items were not used in scoring. This scale was previously applied to Chinese college students and was found to be culturally adaptable [29]. Cronbach’s alpha coefficients were 0.80 and 0.78 for the agency and pathway subscales, respectively, and 0.80 for the full scale. Confirmatory factor analysis (CFA) indicated that the model fit for the hope measure was acceptable: χ^2^/df = 2.27, CFI = 0.97, TLI = 0.96, RMSEA = 0.06.

#### 2.3.2. Social Support

Social support was measured using the Index of Sojourner Social Support (ISSS) developed by Ong and Ward [30]. The scale contains 18 items: 9 assess socioemotional support (e.g., comfort you whenever you feel homesick), and 9 assess instrumental support (e.g., provide necessary information to help orient you to your new surroundings). Each item was rated on a 5-point scale, ranging from 1 (nobody) to 5 (many people). Socioemotional and instrumental support subscale items were summed to yield a total social support score. This scale has been translated into Chinese and has been shown to be culturally adaptable to migrants in China [31]. Cronbach’s alpha coefficients were 0.98 and 0.98 for Han social support and minority social support, respectively. CFA indicated that the model fit was acceptable: χ^2^/df = 2.28, CFI = 0.98, TLI = 0.97, RMSEA = 0.06 (Han social support); χ^2^/df = 2.14., CFI = 0.91, TLI = 0.92, RMSEA = 0.06 (minority social support).

#### 2.3.3. Life Satisfaction

Life satisfaction was measured using the Students Life Satisfaction Scale (SLSS) developed by Huebner [8]. The scale contains seven items; each is rated on a 4-point scale, ranging from 1 (never) to 4 (always). Items 3 and 4 were reverse scored. All items were summed to yield a total hope score. High scores on the SLSS indicate high life satisfaction. This scale has been translated into Chinese and found to be culturally adaptable to college students in China [29]. Cronbach’s alpha coefficient was 0.78; CFA indicated that the model fit was acceptable: χ^2^/df = 2.18, CFI = 0.98, TLI = 0.96, RMSEA = 0.05.

## 3. Statistical Analysis

Statistical analysis was performed by using SPSS 20.0 (IBM., New York, NY, USA) and Mplus 7.1 (Muthén & Muthén, Los Angeles, CA, USA) (Muthén, 1998–2012) [32]. First, descriptive statistics were calculated using SPSS, including means, standard deviations, and bivariate correlations. The distribution of variables in this study was normal. Second, path analysis was used to calculate the effect of hope and social support on life satisfaction and the moderating role of the two types of social support [33]. For hope, two types of social support were specified as exogenous variables. To examine the moderating effect of the two types of social support, two product terms were created by multiplying hope and mean-centered two types of social support (i.e., hope × Han social support) and specified as exogenous variables to predict life satisfaction. The Johnson–Neyman (J-N) statistical analysis was used to further interpret the moderating effect of social support and determine the specific moderating model (enhancing or antagonistic) [34]. Missing data were handled via the full information maximum likelihood method [35]. Third, multigroup analysis was also used to test for statistically significant differences in the path parameters across groups (i.e., male vs. female).

### 3.1. Descriptive Statistics

Table 2 shows the means, standard deviations, and correlations among the major variables.

The results shown in Table 2 indicate that hope was positively associated with life satisfaction (r = 0.46, *p* < 0.01). Han social support was positively associated with hope (r = 0.22, *p* < 0.05) and life satisfaction (r = 0.32, *p* < 0.01). Minority social support was positively associated with life satisfaction (r = 0.14, *p* < 0.05). Han social support was also positively associated with minority social support (r = 0.40, *p* < 0.01). These results infer that moderating variables have small and moderate correlations with dependent and independent variables [36].

### 3.2. Moderating Effect of Social Support on the Relation between Hope and Life Satisfaction

Path analysis was used to calculate the effect of hope and social support on life satisfaction and the moderating role of the two types of social support (Figure 1). Demographic factors (gender, school type, and hometown) were used as control variables.

The model results showed that only Han social support was positively associated with life satisfaction (b = 0.31, β = 0.31 SE = 0.13, *p* < 0.05). Meanwhile, only Han social support moderated the relationship between hope and life satisfaction (b = 0.12 β = 0.12, SE = 0.04, *p* < 0.001). The results indicated that hope was not a promotive factor in life satisfaction among the ethnic minority students, while Han social support was promotive. Furthermore, hope only affected life satisfaction when Han social support existed.

The Johnson–Neyman (J-N) statistical analysis was used to further interpret the moderating effect of Han social support and determine the specific moderating model (enhancing or antagonistic). The J-N analysis indicated that hope was positively associated with life satisfaction at or above the value of 1.56 on Han social support (i.e., −1.55 SD below the mean) (Figure 2). Note that the range for Han social support was [1,5], with the mean as 3.41 and standard deviation (SD) as 1.19. In our sample, 92.3% of participant responses were above this threshold. However, in the range of Han social support below 1.56 (i.e., −1.56 SD below the mean), hope was not associated with life satisfaction. In summary, hope was related to a higher level of life satisfaction with increased Han social support. These results indicated that the specific moderating effect of Han social support was an enhancing model.

### 3.3. Multiple Group Analysis

We used multigroup analysis to examine whether the interaction model differed significantly between the two groups of ethnic minority college students (i.e., female vs. male). We first compared a model in which all paths were freely estimated with a model in which all paths were constrained to be equal. The results showed that no significant difference existed between the two models (Δχ^2^(5) = 1.82, *p* > 0.1), indicating that the interaction model did not differ between the female and male group.

## 4. Discussion

Ethnic minority college students are a disadvantaged group with lower life satisfaction due to acculturative stresses experienced in Han regions [6,7]. Our study focused on exploring the promotive factors and their influencing mechanism on life satisfaction to improve the maladjustment condition of ethnic minority college students.

The promotive effects of hope and two types of social support on life satisfaction were firstly explored. The results showed that only Han social support was positively associated with the life satisfaction of ethnic minority college students in China. This is consistent with previous studies, which likewise found that local social support had promotive effects on adaptation [16,17]. This result enriched the local social support research area by revealing the effect of local social support on well-being in a new cultural group. Our results also found that hope and minority social support did not have promotive effects among ethnic minority college students in China. This aligns with the resilience viewpoint that the effect of protective factors is not always promotive; rather, it can vary according to group, context, and outcome [19].

To explain the influencing mechanism of promotive factors on life satisfaction, the moderating effect of two types of social support in the relation between hope and life satisfaction were further examined based on a protective–protective resilience theoretical framework. The results showed that hope could have a promotive effect in combination with Han social support among both male and female students; namely, increased Han social support can strengthen the promotive effect of hope on life satisfaction. Our results thus provide empirical support for the protective–protective framework of resilience for a different cultural group [19]. Moreover, our results suggest that hope and Han social support should both be strengthened to increase life satisfaction among minority college students in China. The “matching hypothesis” can help explain why hope combined with Han social support, but not with minority social support, can positively affect life satisfaction. The “matching hypothesis” suggests that only the type of social support that meets coping needs can have a buffering effect [37]. In the present study, the function of Han social support as well as participant characteristics could illustrate why only Han social support met the need to increase hope. According to Kim et al., local social support is more important than nonlocal social support for adaptation because it can offer authentic cultural information and resources that nonlocal social support cannot provide [16]. In addition, as newcomers to Han culture, our participants were more in need of Han social support, given their higher levels of confusion and conflict regarding cultural differences compared to upper-level (nonfreshman) students [27]. For these reasons, Han social support, not minority social support, had a buffering effect in the relation between hope and life satisfaction among ethnic minority college students in China, and increased Han social support could strengthen the effect of hope on life satisfaction.

The results of this study not only expanded related empirical research, but also provided a direction for developing culturally competent interventions to improve life satisfaction among ethnic minority college students in China. As the results inferred, increasing Han social support could be the main direction of interventions for its unique effect. We should create warm and accepted circumstances in Han groups, which could make ethnic minority students be willing to seek Han social support and believe the availability. Propagating ethnic minority cultures focusing on the merits and principles in the Han group was also a good way to build accepted circumstance. In addition, increasing group contact could be useful for accepted circumstance, as group contact could effectively decrease outgroup stigma [38]. Finally, we should provide Han social support to ethnic minority college students in an organized and purposeful way to help them grasp cultural knowledge in Han regions. For example, we can organize some lectures on cultural adaptation and assign Han students to help them learn and familiarize the campus environment.

Although the study enriched empirical research of ethnic minority college students and provided a basis for culturally competent intervention, it still has some limitations. First, we tested the moderating role of social support in a cross-sectional study; however, the moderating role can fluctuate at different points in time, as shown in longitudinal research on social support [39,40,41]. Therefore, longitudinal research should be conducted that covers a broader time period (e.g., from enrollment to graduation). Second, more ethnic minority college students in different cities of Han regions should be recruited. Our study only recruited participants in Beijing. As a cosmopolitan, first-tier city in China, Beijing has many specific characteristics that make it different from other third-tier cities that are surrounded by minority populations (e.g., Yin Chuan and Xi Ning). Thus, participants in Beijing cannot solely describe the whole picture of the moderating role of two types of social support. Further research is needed to explore how the relation between hope and life satisfaction is moderated by social support among ethnic minority college students in other cities in China. Third, multimethod and multi-informant studies should be conducted to explore the moderating effect of social support. This study only used the questionnaire method and only collected information on ethnic minority college students in China. Experimental methods and data from other informants, such as teachers and parents, could be used in future studies.

## 5. Conclusions

Based on resilience theory, this study examined the effects of hope and two kinds of social support (Han and minority) on the life satisfaction of ethnic minority college students in China. We also examined the moderating effect of Han social support and minority social support on the relation between hope and life satisfaction. The results indicated that Han social support was promotive factor of life satisfaction. Moreover, the results indicated that Han social support moderated the relation between hope and life satisfaction. Specifically, the moderating role was an enhancing model, indicating that the effect of hope was stronger under the condition of a high level of Han social support. These results extend resilience theory by providing new cultural research–based evidence. Meanwhile, these results also inferred the direction for practical intervention, that is, Han social support should be strengthened to improve life satisfaction among ethnic minority college students in China.

## Figures and Tables

**Figure 1 ijerph-18-02298-f001:**
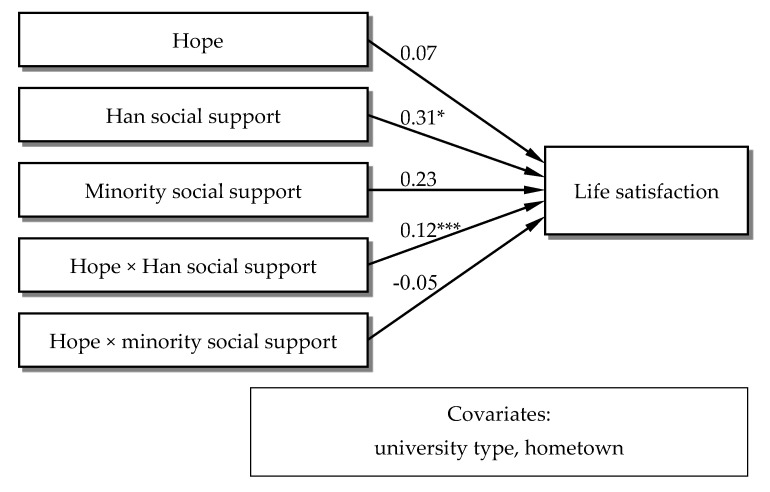
Moderating effect of Han and minority social support. Note: * *p* < 0.05, *** *p* < 0.001.

**Figure 2 ijerph-18-02298-f002:**
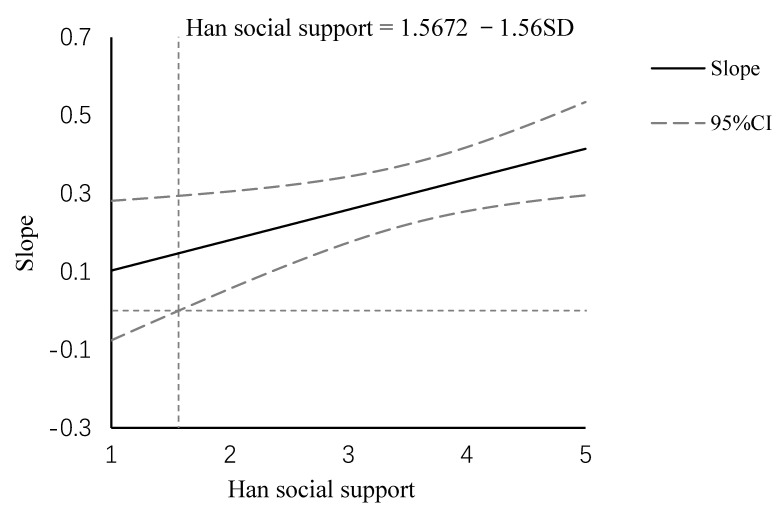
J-N analysis of the moderating effect of Han social support.

**Table 1 ijerph-18-02298-t001:** Sample characteristics (*N* = 362).

Variables	Categorical Values	*n*	%
Gender	male	156	42.8
	female	206	57.2
University type	Ethnic minority college	135	36.8
	Comprehensive college	227	63.2
Hometown	Large city	38	10.6
	Medium-sized city	53	14.8
	Small town	184	50.8
	Village	87	23.8

**Table 2 ijerph-18-02298-t002:** Descriptive statistics and correlations among major variables (*N* = 362).

Variable	M	SD	1	2	3	4
1 Hope	3.58	0.65	—			
2 Han social support	3.25	1.20	0.22 *	—		
3 Minority social support	3.26	1.19	0.09	0.40 **	—	
4 Life satisfaction	2.59	0.54	0.46 **	0.32 **	0.14 *	—

Note: * *p* < 0.05, ** *p* < 0.01.

## Data Availability

Please contact corresponding author to find details regarding data supporting reported results.

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
