# Peer review of "Moderating Roles of Social Support in the Association between Hope and Life Satisfaction among Ethnic Minority College Students in China"

_ijerph, 2021, doi:10.3390/ijerph18052298_

Round 1

Reviewer 1 Report

The paper uses mainly resilience theory to investigate the relations between two types of social support as well as hope on life satisfaction in ethnic minority college students in China based on a sample of N=362 ethnic minority college students in Beijing, China.

While the paper is readable and relevant and to IJERPH, there is room for improvement in several respects:

  1. Title: The manuscript title is too long and hard to understand. “the more the marrier” is unnecessary. “Han” should not be used in title, instead use more generic concepts.
  2. Theoretical background: Acculturation and cultural adaption should be explained more clearly as a starting point.
  3. The relevance of ethnic minority status and its relations to stress, stigma, reduced life satisfaction should be explained and then related to resilience theory
  4. “Han” is not explained at all. It needs to be explained and should be contextualized more generally in the context of local / nonlocal support
  5. I find it confusing and inconsistent to say in the end of the paper that ethnic minority students in China experience almost no discrimination. Such background information needs to be given in the beginning.
  6. Authors should talk about “*ethnic* minority” throughout paper as there are many other minorities (e.g., sexual minorities, gender minorities).
  7. For the key concept “life satisfaction” a definition is given that lacks reference and is factual incorrect: “Life satisfaction refers to the subjective understanding of and level of satisfaction with different *external* conditions in life.” To my understanding life satisfaction is not focused on external conditions exclusively but internal factors play an important role, too (e.g., self concept).
  8. Why did it take 45 minutes (!) to complete this short questionnaire? This is highly unusual and very suspicious and needs clarification.
  9. Authors need to explain the value of the incentive (“10 RMB”) to international readers.
  10. Table 2 formatting needs to improved in terms of alignment to decimals
  11. Figure 1 needs to be improved in terms of symmetry and aesthetics
  12. The authors claim that their results are “providing implications for practical intervention”. However, they need to explain why an intervention would be necessary for ethnic minority college students in China they claim experience almost no discrimination in the first place. Also they need to explain which interventions exactly should be developed based on their results.

Author Response

Dear reviwer,

Reviewer 2 Report

Dear authors. I have found the reading of your original work very interesting. There is not much scientific literature on this field of study, in this particular context, and I really believe that this is a timely and enriching contribution. - It would be advisable to include a clearer recognition of the limitations of the study, especially its representativeness in terms of the general population under study, as well as a discussion of the future research perspectives offered by this study.

Kind Regards

Author Response

Dear reviwer,

 corresponding author

Reviewer 3 Report

I congratulate the authors for the study presented but it requires some profound changes in order to be considered acceptable.

The abstract is not entirely clear, it would be advisable to revise and follow the structure introduction, method, results and conclusions.
In the key words, minority college students in China should be changed to minority college students.
Revise the statement "Some studies have also in-vestigated their influencing mechanisms in relation to life satisfaction [7]". It is stated that they are some studies, when only one citation is incorporated. If the statement is to be maintained, it would be advisable to include more bibliographic citations to justify the idea.

The objectives of the study are as follows:
This study had three aims. First, we explored the relation between hope and life satisfaction among minority college students in China. Second, we investigated the effect of Han social support and minority social support on the life satisfaction of that group. Third, we tested the moderating role of Han social support and minority social support in the relation between hope and life satisfaction.
It is recommended that these be stated in the infinitive, to denote that they are objective.

The study sample is well described but it is necessary to describe the sampling procedure performed. That is, describe the sample selection procedure.
It is necessary to include a section on Analysis, detailing the procedures and statistical analysis performed. This section should indicate whether the distribution of the standard is normal or not, since this will determine the type of correlation analysis performed and presented.

It is indicated that an RMSEA = 0.07 (Social support) is obtained. It is required that the result be justified and revised since it should be equal to or less than 0.06 according to the type of analysis performed.
Regarding the correlation results:
The results shown in Table 2 indicate that hope was positively associated with life 230 satisfaction (r = 0.46, p < 0.01). Han social support was positively associated with hope (r = 231 0.22, p < 0.05) and life satisfaction (r = 0.32, p<0.01). Minority social support was positively 232 associated with life satisfaction (r = 0.14, p < 0.05). Han social support was also positively 233 associated with minority social support (r = 0.40, p < 0.01).
It is indicated that a positive correlation was obtained, but it is necessary to indicate that this correlation is low since the values obtained. 

In order to understand the model, which is supposed to have been calculated with structural equations, it is necessary to present and describe the Parameter estimates of final model (RW = regression weights; SE = standard error; CR = critical radius; SRW = standardized regression weights) produced by the program. In addition, it is requested to see the actual figure produced by the program to verify the results obtained.

Finally, I think it would be advisable to incorporate some other results that give strength to the study. It could be interesting to make a differential for example based on the variables: Gender, University type and Hometown. If some other results are not provided, I consider that the study is poor for the journal. 

Author Response

Dear reviewer,

 corresponding author

Round 2

Reviewer 1 Report

Overall, I think the revision has improved the paper.

I think a final language check by a native speaker of English is necessary.

Reviewer 3 Report

The changes made by the authors are adequate to proceed to publication.